# Integrative Analyses Reveal the Anticancer Mechanisms and Sensitivity Markers of the Next-Generation Hypomethylating Agent NTX-301

**DOI:** 10.3390/cancers15061737

**Published:** 2023-03-13

**Authors:** Byungho Lim, Dabin Yoo, Younghwa Chun, Areum Go, Ji Yeon Kim, Ha Young Lee, Rebecca J. Boohaker, Kyung-Jin Cho, Sunjoo Ahn, Jin Soo Lee, DooYoung Jung, Gildon Choi

**Affiliations:** 1Data Convergence Drug Research Center, Korea Research Institute of Chemical Technology (KRICT), 141 Gajeong-ro, Yuseong-gu, Daejeon 34114, Republic of Korea; 2Pinotbio, Inc., Suwon 16506, Republic of Korea; 3Southern Research, Division of Drug Discovery, Birmingham, AL 35205, USA

**Keywords:** anticancer activity, biomarker, mechanism of action, multiomics, NTX-301

## Abstract

**Simple Summary:**

We thoroughly investigated the experimental and preclinical efficacy of the novel hypomethylating agent (HMA) NTX-301 through comparative analyses with conventional HMAs. By performing multiomics data analyses, we demonstrated the mechanisms of action underlying the anticancer activity of NTX-301 and identified molecular markers that are associated with sensitivity to NTX-301.

**Abstract:**

Epigenetic dysregulation characterized by aberrant DNA hypermethylation is a hallmark of cancer, and it can be targeted by hypomethylating agents (HMAs). Recently, we described the superior therapeutic efficacy of a novel HMA, namely, NTX-301, when used as a monotherapy and in combination with venetoclax in the treatment of acute myeloid leukemia. Following a previous study, we further explored the therapeutic properties of NTX-301 based on experimental investigations and integrative data analyses. Comprehensive sensitivity profiling revealed that NTX-301 primarily exerted anticancer effects against blood cancers and exhibited improved potency against a wide range of solid cancers. Subsequent assays showed that the superior efficacy of NTX-301 depended on its strong effects on cell cycle arrest, apoptosis, and differentiation. Due to its superior efficacy, low doses of NTX-301 achieved sufficiently substantial tumor regression in vivo. Multiomics analyses revealed the mechanisms of action (MoAs) of NTX-301 and linked these MoAs to markers of sensitivity to NTX-301 and to the demethylation activity of NTX-301 with high concordance. In conclusion, our findings provide a rationale for currently ongoing clinical trials of NTX-301 and will help guide the development of novel therapeutic options for cancer patients.

## 1. Introduction

Epigenetic homeostasis is tightly controlled by various epigenetic regulators to maintain normal physiological homeostasis [1]. Increased somatic mutations and transcriptional deregulation in epigenetic regulators impair epigenetic homeostasis, driving tumor development and progression [2]. Epigenetic dysregulation characterized by aberrations in DNA methylation and histone modification is a major hallmark of cancer [3]. Therefore, targeting epigenetic regulators is a well-established therapeutic approach for the treatment of multiple cancers.

DNA methyltransferase I (DNMT1) is an epigenetic ‘writer’ that is responsible for the maintenance of cytosine methylation. DNMT1 transfers a methyl group to the 5′ carbon of cytosine rings in newly synthesized DNA [4], thus ensuring the high-fidelity inheritance of DNA methylation patterns during replication [5]. Emerging evidence suggests that the deregulation of DNMT1 and the resulting aberrant DNA hypermethylation are involved in malignant transformation. Indeed, the reversal of aberrant hypermethylation by inhibiting DNMT1 promoted anticancer activity, especially in hematologic malignancies [6]; this finding partially explains the relevance of DNMT1 as a therapeutic target.

Hypomethylating agents (HMAs) are the most widely used DNMT1 inhibitors. Mechanistically, HMAs, which are cytidine analogs, are incorporated into newly replicated DNA, and DNMT1 becomes sequestered by its formation of covalent complexes with HMAs, eventually resulting in the proteasomal degradation of DNMT1 [7]. The first-generation HMAs decitabine (DAC) and azacitidine (AZA) have been approved for clinical use in the treatment of hematologic malignancies [8]. However, their low response rate, poor bioavailability, and dose-limiting toxicity highlight the need for further treatment improvements [9,10]. The representative effort to overcome the limitations of HMAs is the development of reversible DNMT1-selective inhibitors. GSK3685032, a first-in-class DNMT1-selective noncovalent inhibitor, achieved improved in vivo tolerability, higher antitumor efficacy, and greater DNA demethylation, providing clinical benefits over conventional HMAs [11].

As another effort, we recently described the superior therapeutic potential of NTX-301 (5-aza-4′-thio-2′-deoxycytidine) as a next-generation HMA using six mouse models [12]. This 4′-thio-modified nucleoside analog showed preclinical efficacy, tolerability, and survival outcomes that were superior to those of conventional HMAs when used as a monotherapy or in combination with venetoclax in the treatment of acute myeloid leukemia (AML) [12]. Consistent with our findings, Thottassery et al. showed pharmacological advantages of NTX-301, namely, that NTX-301 exhibited improved chemical stability, DNA incorporation rates, and preclinical activity compared with conventional HMAs [13]. Notably, NTX-301 is better tolerated than DAC, with an at least 10-fold higher selectivity index (ratio of the maximum tolerated dose to the minimal dose required to deplete DNMT1) [13]. Although these findings provide a rationale for the currently ongoing clinical trials that are investigating NTX-301 (NCT04167917, NCT03366116, and NCT04851834), the anticancer activity and mechanisms of action (MoAs) of NTX-301, as well as biomarkers that can be used to predict its efficacy, are not yet elucidated.

Herein, we aimed to thoroughly investigate the experimental/preclinical efficacy of NTX-301 through comparative analyses with the conventional agents DAC and AZA. To understand the widespread therapeutic potential of NTX-301, we determined the sensitivity profiles of 199 cancer cell lines (CCLs) after NTX-301 treatment. By integrating these sensitivity profiles with multiomics data, we investigated the MoAs underlying the anticancer activity of NTX-301 and identified molecular determinants that are associated with sensitivity to NTX-301.

## 2. Materials and Methods

### 2.1. Cell Lines and Reagents

Five human leukemia cell lines (the MV4-11, HL-60, MOLM-13, KG-1, and THP-1 cell lines) and HEK293T cells were cultured in RPMI 1640 and DMEM (Thermo, Waltham, MA, USA), respectively. CRISPR/Cas9 technology was used to knock out *TP53* in MV4-11 cells (Cyagen, Santa Clara, CA, USA). NTX-301 was provided by MercachemSyncom, and DAC, AZA, and mTOR inhibitors (Torin1 and AZD-8055) were purchased from Selleckchem (Houston, TX, USA).

### 2.2. Cell-Based Phenotypic Assays

The sensitivity profiling of 199 CCLs was performed using the OncoPanel™ Multiplex Cytotoxicity Assay (Eurofins, Luxembourg). Briefly, cells grown in RPMI 1640 were seeded into 384-well plates, and NTX-301 was added the following day. NTX-301 was serially diluted and assayed at 10 concentrations with a maximum assay concentration of 0.1% DMSO. After a 3-day incubation, the cells were fixed and stained with nuclear dye to measure cell proliferation by fluorescence intensity. Automated fluorescence microscopy was carried out using a Molecular Devices ImageXpress Micro XL high-content imager, and images were collected with a 4× objective. Sixteen-bit TIFF images were acquired and analyzed with MetaXpress 5.1.0.41 software.

To assess the anticancer effect of NTX-301 at the cellular level, we performed the following assays: (1) A CellTiter-Glo Luminescent Cell Viability Assay (Promega, Madison, WI, USA) was used to determine IC_50_ values. For 2, 4, and 6 days, NTX-301 or DAC was added to 2 × 10^3^ cells plated into 96-well plates in triplicate. Treatment doses ranged from 1.5 nM to 10 µM. (2) For the cell cycle assay, NTX-301 or DAC (500 nM) was added to 5 × 10^5^ MV4-11 cells plated into six-well plates for 2 and 3 days, and the cells were stained with phospho-histone H3 (CST, Danvers, MA, USA) and propidium iodide (Sigma, St. Louis, MO, USA). (3) To assay apoptosis, NTX-301 or DAC (500 nM for 2 and 3 days) was added to 5 × 10^5^ MV4-11 cells plated into six-well plates, and two different methods were used (Annexin V Apoptosis Detection Kit (Thermo) and the Cleaved Caspase-3 Staining Kit (Abcam, Cambridge, UK)). (4) For AML cell differentiation, NTX-301 or DAC (60 and 200 nM) was added to 5 × 10^5^ MV4-11 cells plated into six-well plates for 6 days, and the cells were analyzed with APC-conjugated anti-human CD14 and Alexa Fluor^®^ 488-conjugated anti-human CD11b antibodies (BD Pharmingen™, Franklin Lakes, NJ, USA). The differentiated cell morphology was inspected by Giemsa staining (Thermo). All assays were performed according to the manufacturers’ protocols, and the results were captured using BD FACSCalibur™.

### 2.3. Mouse Study

The mouse study was conducted in accordance with the recommendations of the Guide for Care and Use of Laboratory Animals. All experiments were performed by Charles River Discovery Services (CR Discovery Services, Worcester, MA, USA), which is accredited by the Association for Assessment and Accreditation of Laboratory Animal Care International. A systemic NOD/SCID model bearing luciferase-labeled MV4-11 tumors (n = 8 per group, 4 groups) was established to assess the efficacy of low doses of NTX-301 (0.15, 0.30, and 0.45 mg/kg (p.o.)) in combination with tetrahydrouridine (42.0 mg/kg). Briefly, the mice were administered cyclophosphamide ((100 mg/kg (i.p.) once a day for two days) to ablate their bone marrow three days prior to tumor implantation. After the injection of 5.0 × 10^6^ luciferase-labeled MV4-11 cells, treatment was initiated on day 20 by preparing NTX-301 in N-methyl-2-pyrrolidone (NMP) and then PEG400, followed by the addition of saline with vortexing and sonication. Tumor growth was measured by in vivo bioluminescence imaging on day 42. This study is an unpublished part of a previously reported mouse study [12].

### 2.4. Western Blotting

The cells were lysed with NP-40 buffer (Thermo), and 30 µg of protein was loaded into Mini-PROTEAN precast gels (Bio-Rad, Hercules, CA, USA). Protein transfer was conducted using a Trans-Blot Turbo Transfer System (Bio-Rad). Immunoreactions were detected with SuperSignal™ West Pico or Femto substrates (Thermo) using an iBright System (Thermo). All antibodies were purchased from CST.

### 2.5. Data Analysis

#### 2.5.1. DNA Methylation Analysis

For 2 days, 5 × 10^5^ cells plated into six-well plates in triplicate were treated with NTX-301 or DAC (30 nM for HL-60 and 60 nM for MV4-11). After purifying genomic DNA with a QIAamp DNA Mini Kit (Qiagen, Venlo, The Netherlands), methylome analysis was performed by Macrogen using Illumina Infinium Methylation EPIC BeadChip kits and Illumina GenomeStudio v2011.1 (San Diego, CA, USA). Methylation data points were represented as fluorescent signals of the methylated (M) and unmethylated (U) alleles by subtracting each data point from background intensity calculated from a set of negative controls [14]. The β-value that reflects the methylation level of each CpG site was calculated with the following formula: (max(M, 0))/(|U| + |M| + 100) [14]. To annotate histone modification features at the genome positions of each CpG site, we used H3K27ac, H3K36me3, H3K9me3, and H3K27me3 ChIP-seq data from ENCODE (https://www.encodeproject.org/ (accessed on 24 September 2020)). To annotate replication timing at the genomic locations of each CpG site, we used Repli-seq data from ReplicationDomain (https://www2.replicationdomain.com/ (accessed on 4 November 2020)). ‘Replication timing < 0’, ‘0 ≤ Replication timing < 1’, and ‘1 ≤ Replication timing’ were classified as ‘late’, ‘mid-early’, and ‘early’ replication regions, respectively. The normalized demethylation activity of NTX-301 (versus DAC) was calculated as follows: (the number of CpGs within a given genomic feature that were demethylated by NTX-301)/(the number of CpGs within a given genomic feature that were demethylated by DAC).

#### 2.5.2. Transcriptome Analysis

A total of 5 × 10^5^ cells were plated into six-well plates the day before drug treatment, and NTX-301 or DAC (60 nM for MV4-11, 30 nM for HL-60, and 15 nM for MOLM-13) was administered in triplicate for 2 days (2 and 4 days for MV4-11). After extracting total RNA using an RNeasy mini kit (Qiagen), RNA purity and integrity were assessed by an ND-1000 spectrophotometer. Transcriptome data were produced using Illumina RNA sequencing (for HL-60 and MOLM-13) and Affymetrix GeneChip^®^ Human Gene 2.0 ST Array (for MV4-11). Sequencing libraries were prepared using a TruSeq Stranded Total RNA LT Sample Prep Kit (Gold), and paired-end sequencing was conducted by Macrogen using the Illumina platform. After completing quality control using FastQC, sequencing reads were preprocessed by trimming with Trimmomatic 0.38 [15]. Sequencing reads were then mapped with HISAT2 [16], and transcript assembly and quantification were conducted using StringTie [17]. We used Enrichr [18], GSEA [19,20], IPA (Qiagen), and Gene Ontology analyses to characterize the biological pathways associated with gene sets.

#### 2.5.3. Integrative Data Analysis

To perform integrative multiomics analyses across cancer cell lines, we downloaded Cancer Cell Line Encyclopedia (CCLE) datasets including somatic mutations, gene expression, RPPA, and DNA methylation from cBioPortal (http://www.cbioportal.org/study/clinicalData?id=ccle_broad_2019 (accessed on 6 August 2020)). To identify biomarkers associated with sensitivity to NTX-301, we classified 199 CCLs (profiled for sensitivity to NTX-301) into responders and nonresponders based on their AUC values (bottom third vs. top third). Then, we statistically examined differentially expressed genes, differential mutational events, and differentially methylated regions between the responders and nonresponders. We also integrated IC_50_ values with RPPA data from CCLE to identify IC_50_-correlated proteins in 29 blood CCLs.

### 2.6. Statistical Analysis

GraphPad Prism v9.0 (GraphPad Software) was used for statistical analyses and graphical presentation. The two-tailed unpaired t test was used to compare the mean differences between the two groups. To analyze overall survival, we generated Kaplan–Meier curves and performed the log-rank (Mantel–Cox) test. The synergistic effect of drug combinations was determined by the combination index (CI) using CompuSyn. The experiments were performed in at least triplicate, and statistical significance was defined as *p* < 0.05.

### 2.7. Data Availability

All data are available from the NCBI GEO Datasets (https://www.ncbi.nlm.nih.gov/geo/ (accessed on 10 October 2019)) via the accession numbers GSE188392, GSE187285, and GSE187293.

## 3. Results

### 3.1. Sensitivity Profiling of NTX-301 in 199 CCLs

In a recent companion paper, we clearly demonstrated that NTX-301 exhibited better therapeutic potential than conventional HMAs in preclinical models of AML [12]. However, the applicability of NTX-301 as a broad-spectrum anticancer agent has not yet been evaluated. To comprehensively assess the anticancer activity of NTX-301, we evaluated the viability of 199 CCLs after NTX-301 treatment based on two sensitivity metrics: the IC_50_ value and the area under the dose–response curve (AUC) (Appendix A). Consistent with the current use of HMAs as therapeutics for treating hematologic malignancies, NTX-301 displayed skewed sensitivity toward CCLs of hematopoietic origin (Figure 1A; odds ratio (OR)_NTX-301_ = 3.97, *p* = 0.0003). When analyzed with public data that are available through CTRP [21], DAC and AZA also showed the highest anticancer activity in hematopoietic CCLs (Figure 1B,C). The absolute cytotoxic activity of DAC (AUC_DAC_: 2.1–18.1) was much stronger than that of AZA (AUC_AZA_: 10.9–18.7), while that of NTX-301 could not be directly compared due to the use of different experimental platforms.

NTX-301 and DAC exhibited relatively low efficacy against CCLs that originated from the breast, skin, and CNS (OR < 1), but these drugs exerted opposite effects against kidney-derived CCLs (OR_NTX-301_ = 3.20 vs. OR_DAC_ = 0.27) (Figure 1A,B). We then compared the relative efficacy of NTX-301 and DAC against solid CCLs. NTX-301 treatment caused stronger cytotoxicity in 34 of 160 solid CCLs (21.3%) compared with the median AUC_NTX-301_ observed in hematopoietic CCLs. However, DAC achieved higher efficacy in only 6 of 668 solid CCLs (0.89%) compared with the median AUC_DAC_ observed in hematopoietic CCLs. Thus, this sensitivity profiling revealed that NTX-301 primarily sensitizes hematologic malignancies and that it exhibits superior and broader anticancer activity against solid cancers.

### 3.2. Cellular Phenotypes Associated with the Antileukemic Activity of NTX-301

To dissect the primary activity of NTX-301 in hematopoietic CCLs, we assessed cellular phenotypic changes that occurred after NTX-301 treatment in detail. Cell viability assays revealed that NTX-301 exerted stronger effects than DAC on all five AML cell lines that were examined (Figure 2A). Even at early treatment stages (2 days), NTX-301 promoted stronger cytotoxicity than DAC (Figure 2A). Given the slow kinetics of demethylation, this rapid effect of NTX-301 may suggest that its effect is mediated by nonepigenetic mechanisms. Both agents tended to be more effective in p53-wild-type (WT) AML cells (MV4-11 and MOLM-13) than in p53-null AML cells (HL-60, THP-1, and KG-1) (Figure 2A). However, when applied for a long period of time (6 days), the potency of both agents was dramatically improved, regardless of p53 status (Figure 2A), suggesting the anticancer effects of HMAs through passive demethylation after multiple rounds of replication.

Consistent with the early-onset effect upon NTX-301 treatment, cell cycle analysis revealed that NTX-301 increased S-phase arrest (30% with NTX-301 vs. 7% with DAC) and decreased mitotic progression more significantly than DAC after 2 days of treatment (Figure 2B). Moreover, apoptosis analyses with two different methods (Annexin V staining and cleaved caspase-3) consistently showed that NTX-301 increased the apoptotic cell population more than DAC after 2 days of treatment (Figure 2C,D). Previously, long-term treatment with HMAs at low doses resulted in p53-independent anticancer effects dominated by AML differentiation rather than by cytotoxicity [22]. Hence, we finally performed an AML differentiation assay and found that a low dose of NTX-301 (60 nM) increased the population of CD11b^+^/CD14^+^ differentiated cells more markedly than DAC (Figure 2E). The change was accompanied by differentiation-like morphological changes (increased cell size, cytoplasm vacuolization, and decreased nuclear–cytoplasmic ratio; Figure 2F). Collectively, these results suggested that the superior antileukemic activity of NTX-301 was attributed to its more effective induction of cell cycle arrest, apoptosis, and differentiation.

### 3.3. Antitumor Efficacy of NTX-301 in a Preclinical Model of AML

Given its superior efficacy in vitro, low doses of NTX-301 could achieve a profound therapeutic effect. Recently, even a lower dose of NTX-301 alone (2.0 mg/kg (p.o.)) resulted in greater efficacy and better survival outcomes than treatment with AZA alone (5.0 mg/kg (i.p.)) or the combination of AZA (2.5 mg/kg (i.p.)) + venetoclax (50–100 mg/kg (p.o.)) [12]. As a follow-up study, we sought to test the efficacy of low doses of NTX-301 in vivo using the same model as that in the previous experiment [12], namely, a systemic NOD/SCID mouse model that bears luciferase-labeled MV4-11 cell-derived tumors. NTX-301 is a good substrate for cytidine deaminase, which catalyzes the conversion of deoxycytidines into deoxyuridines [10]. Therefore, to prevent the clearance of NTX-301, we administered the cytidine deaminase inhibitor tetrahydrouridine (42.0 mg/kg) 1 h before NTX-301 treatment. After NTX-301 treatment at doses of 0.15, 0.30, and 0.45 mg/kg (p.o.), which were 13.3, 6.7, and 4.4 times lower than the previously used dose (2.0 mg/kg) [12], respectively, bioluminescence imaging showed substantial in vivo tumor regression (Figure 3A,B). These doses showed weaker activity than 2.0 mg/kg (p.o.) NTX-301, but the effects were comparable to those of 5.0 mg/kg (i.p.) AZA (Appendix A), and no notable weight loss was observed (Figure 3C).

### 3.4. Transcriptome Analyses Cataloged the MoAs of NTX-301

Clinical responses to HMAs are frequently not apparent for up to six treatment cycles [23]. Given that HMAs are currently being used to treat elderly patients with AML who may not be able to tolerate long-term treatment, predicting the response to HMAs is important for avoiding unnecessary treatment. Hence, we attempted to identify markers of sensitivity to NTX-301, which also participate in the MoAs of NTX-301, through integrative multiomics analyses.

First, to elucidate the MoAs of NTX-301, we examined the biological features of NTX-301-induced transcriptome alterations in the three AML CCLs (MV4-11, MOLM-13, and HL-60) using RNA sequencing. In support of the superior efficacy of NTX-301, heatmap analysis showed that NTX-301 elicited more intense and extensive transcriptional changes than DAC (Figure 4A). Gene set enrichment analysis (GSEA) revealed that the gene sets that were more strongly upregulated by NTX-301 than by DAC were associated with biological processes such as the DNA damage response (DDR), the p53 pathway, the immune response, and apoptosis (Figure 4B–D). In our previous study, the DDR and the p53 pathway were already described as the most significantly activated MoAs upon NTX-301 treatment [12]. Gene Ontology analysis also highlighted the enrichment of the DDR, the p53 pathway, and immune responses as activated hallmarks (Appendix A), indicating their importance as MoAs. In addition, NTX-301 reversed immunotherapy resistance signatures [24] more significantly than DAC (Appendix A), indicating its potential to be used in combination with immunotherapy [25]. Twenty genes that were commonly upregulated by NTX-301 in the three different CCLs were involved in immune activation, and these genes included known tumor suppressors such as *BTG2* [26], *ALOX5* [27], *SOCS1* [28], and *TP53INP1* [29] (Appendix A).

GSEA querying genes that were more strongly downregulated by NTX-301 implied the suppression of biological hallmarks, including cholesterol biosynthesis, the cell cycle, DNA replication, and pyrimidine metabolism (Figure 4B–D). Ingenuity Pathway Analysis (IPA) identified SREBF1/2 and mTORC1/2, which are master regulators of cholesterol synthesis and lipid metabolism, as upstream regulators of these downregulated genes (Appendix A). Notably, the downregulation of cholesterol biosynthesis-related genes was associated with enhanced overall survival in four AML cohorts (Appendix A), suggesting a clinical benefit of this MoA. NTX-301 also suppressed the oncogenic Myc and E2F signatures more strongly than DAC, which partially explained its higher efficacy (Appendix A). Moreover, NTX-301 repressed a transcriptional program associated with oxidative phosphorylation, which is a process that is essential for the maintenance and survival of leukemic stem cells [30], more significantly than DAC (Appendix A). NTX-301 also suppressed the transcription of pyrimidine metabolism-related genes (Figure 4B,C), and it is the known MoA of conventional HMAs [31].

### 3.5. Integrative Data Analyses Identified Transcriptional Events Associated with Sensitivity to NTX-301

Next, to identify molecular events associated with sensitivity to NTX-301, we performed integrative analyses of the sensitivity profiles of 199 CCLs and multiomics data from the Cancer Cell Line Encyclopedia (CCLE). The 199 CCLs were classified into responders and nonresponders based on their AUC values (bottom third vs. top third), and CCLs with intermediate AUC values were excluded. To account for a confounding effect of tissue type, we included a binary covariate denoting whether a cell line was of hematopoietic origin. Then, we investigated transcriptional events associated with sensitivity to NTX-301 by examining differentially expressed genes between the responders and nonresponders.

As a proof of concept, our analysis reproduced a previous finding related to conventional HMAs, namely, that the efficacy of HMA is correlated with the expression levels of the HMA metabolism-related genes *DCK*, *CDA*, and *SAMHD1* (Appendix A) [31,32]. In addition, our analysis comprehensively identified 362 sensitivity-associated and 325 resistance-associated genes (P_adj_ < 0.05; Figure 5A, Appendix A). Subsequent GSEA revealed that the p53 pathway and apoptosis are sensitivity-associated processes and that the mTOR pathway is a resistance-associated process (Figure 5B). Similarly, an integrative analysis of IC_50_ values from 29 blood CCLs and reverse-phase protein array (RPPA) data from CCLE revealed that sensitive CCLs (low IC_50_) tended to exhibit higher expression levels of apoptotic proteins (e.g., cleaved PARP (r = −0.30) and Bak (r = −0.38)), whereas resistant CCLs (high IC_50_) tended to exhibit higher expression levels of the antiapoptotic protein Bcl-xL (r = 0.35) and the mTOR pathway-associated proteins phosphorylated-mTOR (r = 0.37) and phosphorylated-S6 (r = 0.42) (Figure 5C).

Our analyses repeatedly identified the p53 pathway as a molecular event that is associated with sensitivity to NTX-301 as well as the MoA of NTX-301. To validate the impact of the pathway on sensitivity to NTX-301, we established *TP53*-knockout (KO) MV4-11 cells using CRISPR-Cas9. Indeed, *TP53* KO dramatically decreased the efficacy of NTX-301, resulting in a >14-fold increase in IC_50_ (Figure 5D). We also noted that the resistance-associated mTOR pathway was essential for the tumorigenesis and propagation of AML as well as various solid tumors [33]. To confirm the impact of the mTOR pathway on resistance, we examined the efficacy of NTX-301 when used in combination with the mTOR inhibitors Torin1 and AZD-8055. As expected, the combination of the inhibitors synergistically improved the efficacy of NTX-301 (Figure 5E). Accordingly, the integrative data analyses revealed intrinsic mechanisms related to sensitivity/resistance to NTX-301.

### 3.6. Integrative Data Analyses Identified Mutation Events That Are Associated with Sensitivity to NTX-301

We next examined the somatic mutations associated with sensitivity to NTX-301. To exclude clinically irrelevant mutations, we filtered out mutations that are predicted to be passenger mutations and are not present in patient-derived tumors (TCGA data). The examination of differential mutational shifts between the responders and nonresponders showed that the responders tended to have a higher mutational burden than the nonresponders (Figure 6A). Indeed, AUC-based sensitivity values were significantly correlated with the number of mutations identified in CCLs (Figure 6B). A high mutational burden can be caused by the hyperproliferative features of cancer cells that bring about excessive replication stress and spontaneous DNA damage [34]. In fact, the responders tended to have shorter doubling times (Figure 6C) and tended to exhibit higher expression levels of pChk1, pRb, Myc, and FOXM1, which are required for hyperproliferation (Figure 5C). These results may support the hypothesis that NTX-301 more effectively sensitizes hyperproliferating cells because hyperproliferating cells rapidly incorporate NTX-301 during replication.

Due to this mutational bias in the responders, we focused only on mutations that frequently occurred in the nonresponders and identified mutations in three genes: *TP53*, *RB1*, and *NSD1* (Figure 6A). Consistent with the MoAs of NTX-301, the nonresponders showed a higher frequency of mutations in *TP53* (OR = 0.41, *p* = 0.035) and *RB1* (OR = 0.24, *p* = 0.044) than the responders. The *NSD1* mutations that frequently occurred in the nonresponders (OR = 0.098, *p* = 0.017) were also noteworthy because *NSD1* mutations have been demonstrated to trigger global hypomethylation [35,36], which may affect the efficacy of HMAs [37].

### 3.7. Integrative Data Analyses Identified Epigenomic Events That Are Associated with Sensitivity to NTX-301

We then examined differential methylation patterns in the promoter regions of the responders versus nonresponders and found a substantial shift toward global hypermethylation in the responders (Figure 6D). By examining the 352 most differentially methylated regions (P_adj_ < 0.05; hereafter abbreviated as DMR352; Appendix A), we found that the average level of DMR352 methylation showed a significant correlation with the log_2_-transformed IC_50_ value (Figure 6E). The significance of DMR352 far exceeded that of 100,000 iterations of correlations calculated from 352 regions that were randomly selected from 21,337 promoter regions (Figure 6F); these results indicated the importance of DMR352 as a sensitivity marker. DMR352 may have a functional relationship with HMA, as it significantly overlapped with the upstream promoter regions of genes whose expression is regulated by DAC (Appendix A).

The association of global hypermethylation with sensitivity led us to determine whether NTX-301 functions as an HMA. To decipher the influence of NTX-301 on the methylome, we investigated genome-wide changes in methylation in two CCLs (MV4-11 and HL-60) using an 850 K methylation array. Consistent with a prior finding that NTX-301 and DAC completely deplete DNMT1 [12], both agents promoted significant global demethylation (Figure 6G). However, despite their similar DNMT1-depleting activities, the demethylation activity of NTX-301 was unexpectedly weaker than that of DAC (Figure 6G). The examination of differentially methylated CpGs (DMCs) also showed that cells treated with these two agents shared many DMCs, but NTX-301-induced DMCs were subordinate to DAC-induced DMCs (Figure 6H).

We hypothesized that the distinct demethylation activities of these two drugs might be attributed to their differential activity in inducing S-phase arrest (Figure 2B), since the DNMT1-mediated maintenance of methylation depends on DNA replication [38]. Therefore, we explored the degree of demethylation during the progression of replication using Repli-seq data. Surprisingly, the CpGs that were demethylated by NTX-301 mapped predominantly to genomic regions that were replicated during the early S-phase (early replication timing) in MV4-11 cells, which were arrested in the S-phase (Figure 7A). Moreover, the changes in demethylation (∆β) induced by NTX-301, but not those induced by DAC, displayed a marked inverse correlation with replication timing (Figure 7B). The effect was greatly diminished in HL-60 cells, which were not arrested in the S-phase, although NTX-301 still led to a slight skewing of demethylation toward early replication (Figure 7A). These results suggest that the intra-S-phase arrest induced by NTX-301 allows for preferential demethylation during early replication and prevents replication-coupled passive demethylation.

To further compare the demethylation activities of NTX-301 and DAC in various genomic contexts, we analyzed methylome data from HL-60 cells, which do not exhibit notable S-phase arrest and are thus sensitive to NTX-301-induced demethylation. The normalization of the number of DMCs that are demethylated by NTX-301 (vs. DAC) showed that the demethylation activity of NTX-301 was weaker than that of DAC in CpG islands but stronger than that of DAC in promoter regions (TSS1500) (Figure 7C). The demethylation activities of both agents were comparable in gene bodies and noncoding regions (Figure 7C). Since DNA replication is spatially and temporally coordinated with chromatin organization [39], we also evaluated the demethylation of chromatin features annotated using ChIP-seq data. The demethylation activity of NTX-301 was considerably lower than that of DAC in H3K9me3-marked regions, which are associated with heterochromatin and late replication timing, but the activity of NTX-301 was higher than that of DAC in H3K27ac-marked regions that are associated with open chromatin and active gene regulation (Figure 7D). Overall, our findings suggested that basal methylation levels had value for predicting NTX-301 sensitivity, while on-treatment demethylation levels may not properly reflect the efficacy of NTX-301 due to its unique demethylation patterns.

## 4. Discussion

This study demonstrated previously unrecognized characteristics of NTX-301 through comparative analyses with conventional HMAs. Experimental investigations revealed that NTX-301 exerts primary anticancer effects on blood cancers, while it exhibits improved sensitivity profiles in solid cancers; additionally, integrative multiomics analyses revealed the MoAs and demethylation activity of NTX-301 as well as markers of sensitivity to NTX-301. Despite minor differences in chemical structures, NTX-301 exhibited substantially improved therapeutic potential at the molecular and cellular levels compared with conventional HMAs. In particular, low doses of NTX-301 (0.15, 0.30, and 0.45 mg/kg (p.o.)) achieved profound antitumor efficacy in vivo, showing comparable effects to a high dose of AZA (5.0 mg/kg (i.p.)). Improved anticancer effects when applied at low concentrations for a long period of time (Figure 2A) may indicate that NTX-301 has an anticancer effect through the involvement of passive demethylation after multiple rounds of replication under the depletion of DNMT1. These findings also suggest a therapeutic advantage of NTX-301 in overcoming toxicity when applied with conventional HMAs at high concentrations. According to our study and other studies [10,12], NTX-301 may satisfy many of the requirements for next-generation HMAs: (1) improved efficacy, (2) better survival benefits, (3) lower toxicity, (4) improved chemical stability and oral bioavailability, and (5) easy activation into triphosphate and incorporation into DNA without replication termination. Nonetheless, in-depth studies are still needed to address how NTX-301 is metabolized in cells and what off-target activity the metabolites possess.

Cellular and molecular analyses revealed MoAs underlying the superior efficacy of NTX-301. In addition to its reported role in promoting stronger transcriptional reprogramming toward normal myeloid-like signatures [12], NTX-301 promoted more robust and extensive changes in the transcription of antileukemic hallmark genes than DAC. In particular, the DDR-p53 pathway may be an essential MoA, as transcriptome analyses repeatedly pinpointed this pathway, and p53 KO dramatically decreased the efficacy of NTX-301. The pathway acts as an antileukemic barrier by enforcing DDR-induced AML cell differentiation [40]. While the exact mechanisms by which NTX-301 predominantly activates the DDR are unknown, two possible mechanisms can be suggested. The first possible mechanism is that the bulky DNMT1 adducts that are trapped in NTX-301-substituted DNA can obstruct oncoming replication forks, causing replication fork collapse and DNA double-strand breaks [41]. The second possible mechanism involves the disruption of pyrimidine metabolism. Unbalanced dNTP pools cause the misincorporation of nucleotides and induce DDR [42]. In fact, our transcriptome analysis suggested that pyrimidine metabolism was inhibited after NTX-301 treatment (Figure 4B,C).

The induction of immune responses, which is another important MoA that was elucidated by transcriptome analyses, not only reflects NTX-301-induced AML cell differentiation but may also represent epigenetic immunosensitization [43]. Recently, HMAs were found to trigger type I interferon signaling via double-stranded RNA and upregulate surface antigens, viral defense pathway components, and PD-L1 [44]. Therefore, the effect of NTX-301 when used in combination with immunotherapy is an interesting area to investigate. The inhibition of cholesterol biosynthesis-related genes is also another notable MoA because cholesterol starvation was previously shown to induce AML cell differentiation [45], and cholesterol synthesis inhibitors sensitized AML to standard antileukemic regimens by blocking adaptive cholesterol responses [46,47].

Our findings suggested that NTX-301 may be applicable to the treatment of solid tumors as well as hematologic malignancies. Among the 199 CCLs, the potency of NTX-301 against solid CCLs greatly surpassed that of conventional HMAs. Notably, NTX-301 showed promising preliminary efficacy against advanced solid tumors (NCT03366116), achieving stable disease in 11 of 14 patients (disease control rate = 78.6%). Although none of the conventional HMAs have been approved for the treatment of solid tumors, this observation provides a rationale for a current phase I trial that is evaluating the safety and tolerability of NTX-301 when used in combination with platinum-based chemotherapies for the treatment of ovarian and bladder cancer (NCT04851834).

Our biomarker analyses revealed various molecular events that are associated with sensitivity to NTX-301. Although these results are limited due to a lack of clinical validation, they provide insights into the important aspects of NTX-301 treatment: (1) the role of global hypermethylation or DMR352 as a predictor of sensitivity, (2) the identity of NTX-301 as an HMA that more effectively sensitizes CCLs with aberrant hypermethylation, (3) the high concordance between markers of sensitivity and MoAs, and (4) the role of the mTOR pathway as an intrinsic mechanism underlying resistance to NTX-301. We therefore anticipate that these biomarkers will be validated in clinical trials.

## 5. Conclusions

Our study provides a rationale for the further development of NTX-301 for use in the clinic. We believe that the MoAs and biomarkers identified in this work will improve our understanding of the potential of NTX-301 to function as a next-generation HMA and facilitate patient stratification during the development of NTX-301, thus providing a novel therapeutic option for cancer patients.

## Figures and Tables

**Figure 1 cancers-15-01737-f001:**
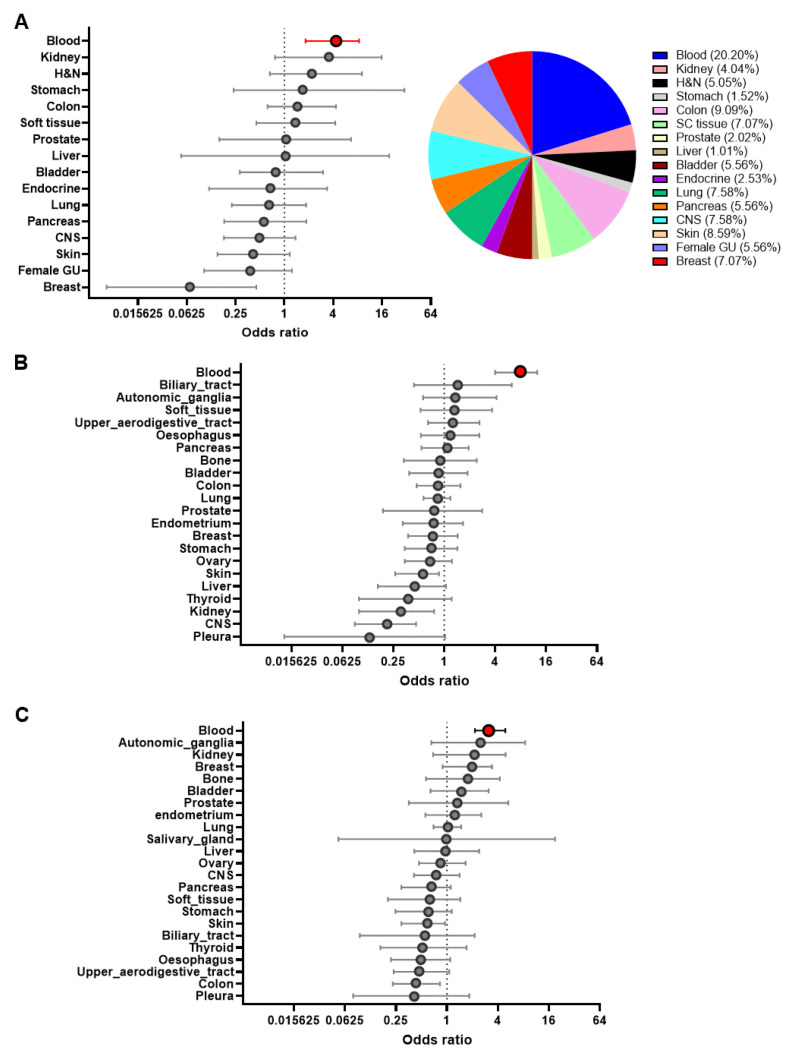
Comprehensive sensitivity profiles of NTX-301 and conventional HMAs in cancer cell lines (CCLs). (**A**–**C**) Plots of odds ratios (ORs) and 95% confidence intervals of CCL sensitivity to NTX-301 (**A**, left), DAC (**B**), or AZA (**C**) grouped according to tissue type. Sensitive CCLs were defined as those with the lowest 50% AUC values of all the CCLs that were examined. The pie chart shows the proportion of each cancer type among the 199 CCLs (**A**, right). AUC values for DAC and AZA are available from the Cancer Therapeutics Response Portal (CTRP). H&N, head and neck; CNS, central nervous system; GU, genitourinary system. Blood cancer is highlighted with red circles.

**Figure 2 cancers-15-01737-f002:**
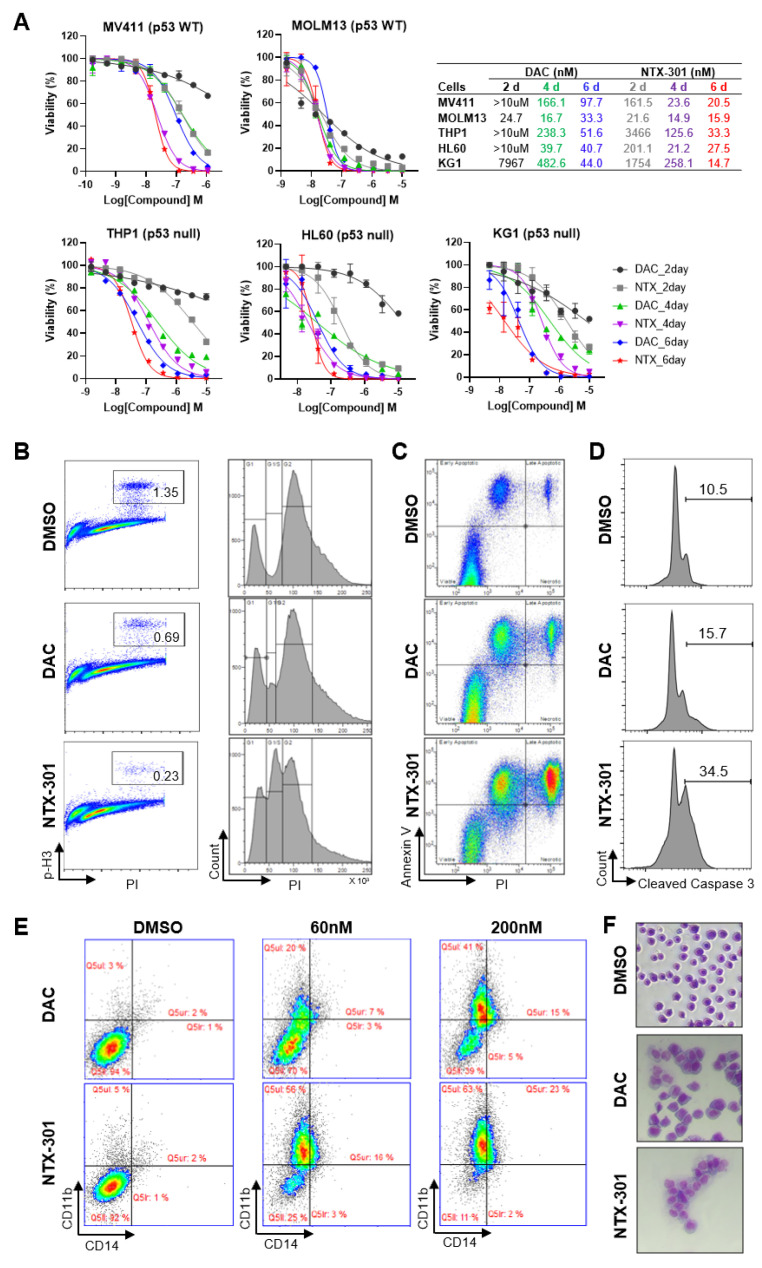
Cell-based phenotypic assays revealed the superior antileukemic activity of NTX-301. (**A**), Dose–response curves of five AML cell lines (MV4-11, MOLM-13, HL-60, THP-1, and KG-1) that were treated with NTX-301 or DAC for the indicated times. The p53 status of the cell lines is shown. (**B**–**E**), Flow cytometric analyses of MV4-11 cells that were treated with NTX-301 or DAC (500 nM) for 2 days to evaluate cell cycle progression by phospho-histone H3 and PI staining (**B**), apoptosis by Annexin V and PI staining (**C**), and apoptosis by cleaved Caspase 3 and PI staining (**D**), and those treated with NTX-301 or DAC (60 and 200 nM) for 6 days to evaluate AML differentiation by CD14 and CD11b staining (**E**). (**F**), A representative micrograph showing the morphology of MV4-11 cells that were treated with NTX-301 or DAC (60 nM).

**Figure 3 cancers-15-01737-f003:**
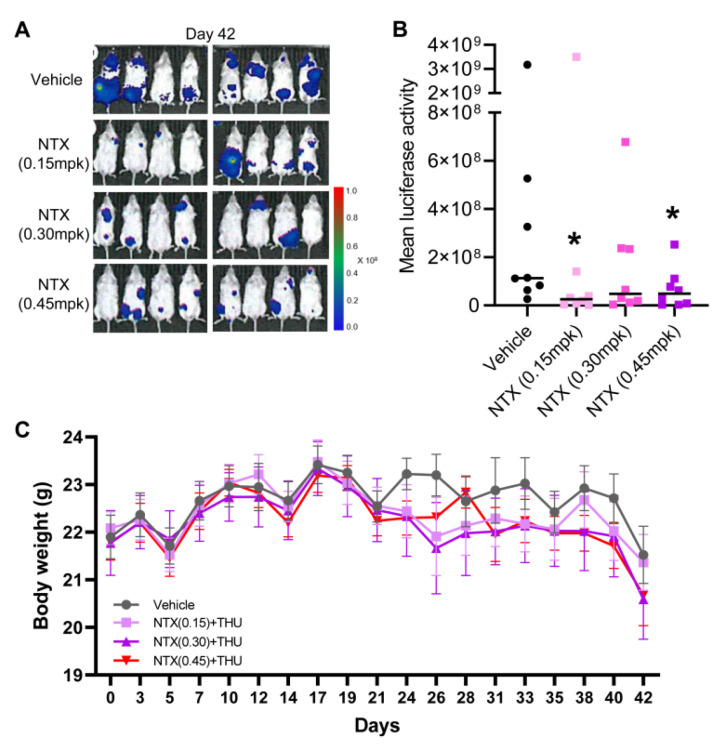
The antitumor efficacy of low doses of NTX-301 in a preclinical model of AML. (**A**–**C**), Bioluminescence images of tumor growth (**A**), quantitative measurements of the bioluminescence emission (photons/sec) (**B**), and body weight (**C**) of NOD/SCID mice (n = 8 per group) that were transplanted with luciferase-labeled MV4-11 cells and treated with NTX-301 (p.o., daily at 0.15, 0.30, or 0.45 mg/kg (mpk) in combination with tetrahydrouridine (THU, 42.0 mg/kg)) on day 42. *, *p* < 0.05.

**Figure 4 cancers-15-01737-f004:**
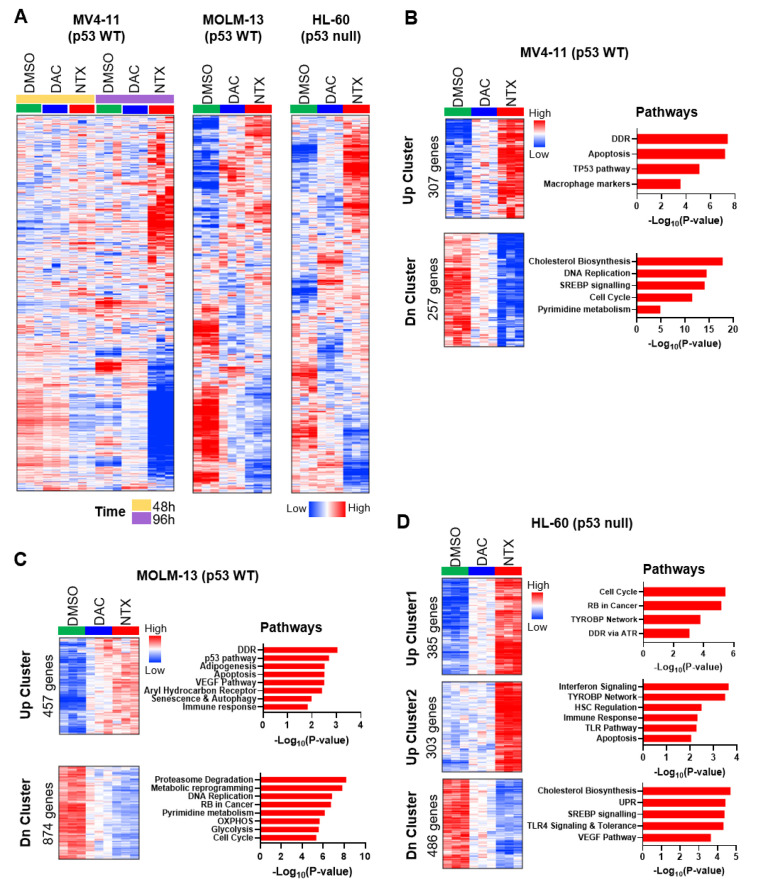
Transcriptome analysis revealed the MoA of NTX-301. (**A**), Heatmaps displaying global changes in the transcriptomes of MV4-11, MOLM-13, and HL-60 cells after treatment with DMSO, NTX-301, or DAC (60 nM for MV4-11, 30 nM for HL-60, and 15 nM for MOLM-13). (**B**–**D**), Heatmaps showing gene sets that were more strongly up- (Up Cluster) or downregulated (Dn Cluster) by NTX-301 than by DAC and bar plots showing biological pathways associated with each gene set in MV4-11 (**B**), MOLM-13 (**C**), and HL-60 (**D**) cells. (**B**–**D**) are updated figures of Appendix A, previously published in [12].

**Figure 5 cancers-15-01737-f005:**
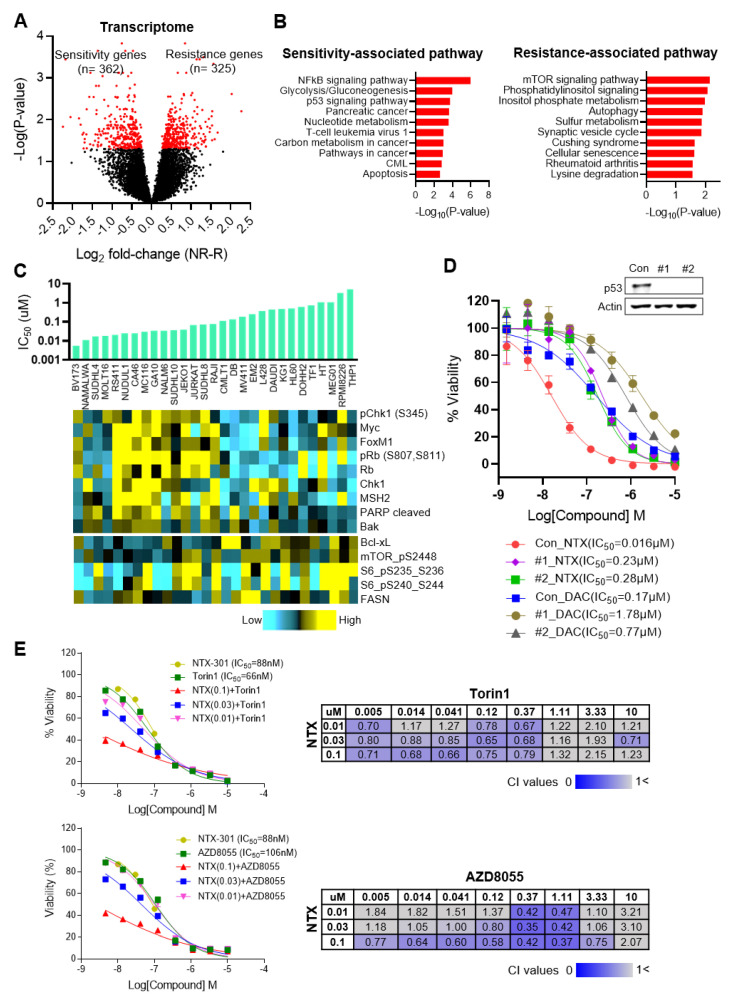
Integrative analyses using 199 CCLs identified transcriptional events that are associated with sensitivity to NTX-301. (**A**), A volcano plot showing transcriptional events that are associated with sensitivity to NTX-301. On the X-axis, ‘log_2_ fold-change(NR-R)’ indicates the log_2_-transformed fold change in the expression of each gene in the nonresponders (NR) relative to the responders (R). The red dots highlight significant events that were identified by the cutoff criteria of *p* < 0.05. (**B**), Bar plots showing biological pathways that are significantly associated with sensitivity or resistance to NTX-301 at the transcriptome level. (**C**), Image integrating IC_50_ values upon NTX-301 treatment (bar plot) and protein expression levels from the Cancer Cell Line Encyclopedia (CCLE) RPPA data (heatmap) in 29 blood CCLs. (**D**), Dose–response curves of MV4-11 parental (con) and p53-KO cell lines (#1, #2) after treatment with NTX-301 or DAC for 3 days. p53 KO was confirmed by western blotting. (**E**), IC_50_ curves of NTX-301 and the mTOR inhibitors Torin1 (top) and AZD-8055 (bottom), used as monotherapy or in combination (left). Matrices show the combination index (CI) values of NTX-301 + mTOR inhibitors at the indicated concentrations in MV4-11 cells (right). CI values < 1 indicate synergism (blue), and CI values > 1 indicate no synergism (gray).

**Figure 6 cancers-15-01737-f006:**
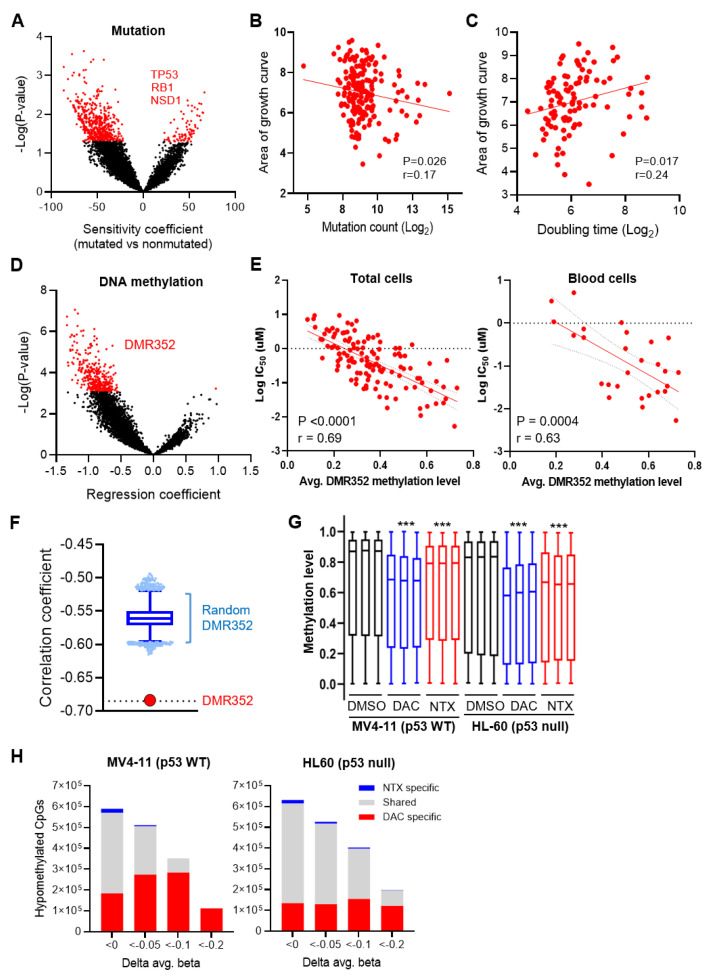
Integrative analyses using 199 CCLs identified mutational and epigenomic events that are associated with sensitivity to NTX-301. (**A**), A volcano plot showing mutational events that are associated with sensitivity to NTX-301. On the X-axis, negative coefficient values indicate the skewing of each event toward the responders relative to the nonresponders. The red dots highlight significant events that were identified by the cutoff criteria of *p* < 0.05. (**B**,**C**), Scatter plots showing the log_2_-transformed mutation count (**B**) and doubling time (**C**) of CCLs plotted against the corresponding AUC values after NTX-301 treatment. (**D**), A volcano plot showing DNA methylation events that are associated with sensitivity to NTX-301. On the X-axis, negative coefficient values indicate the skewing of each event toward the responders relative to the nonresponders. The red dots highlight significant events (DMR352) that were identified by the cutoff criteria of a false discovery rate (FDR) < 0.05. (**E**), Scatter plots showing correlations between the average methylation level of DMR352 and the log-transformed IC_50_ value in all of the CCLs (left) and blood CCLs (right). *p* values and correlation coefficients (r) are shown. (**F**), Plot comparing correlation coefficients between the defined DMR352 and a random DMR352 (100,000 iterations of 352 randomly selected regions). (**G**), Box plots showing the distribution of levels of methylation (β-values) in 862,927 CpGs in triplicate samples of MV4-11 and HL-60 cells treated with NTX-301 or DAC (60 nM for MV4-11 and 30 nM for HL-60). ***, *p* < 0.0001. (**H**), Bar plots showing the number of demethylated CpGs among 862,927 CpGs in MV4-11 (left) and HL-60 (right) cells after treatment with NTX-301 or DAC. Demethylation was defined by the criteria of *p* < 0.05 and ∆β < 0, <−0.05, <−0.1, and <−0.2. ‘Shared’ indicates CpGs that were demethylated by both NTX-301 and DAC.

**Figure 7 cancers-15-01737-f007:**
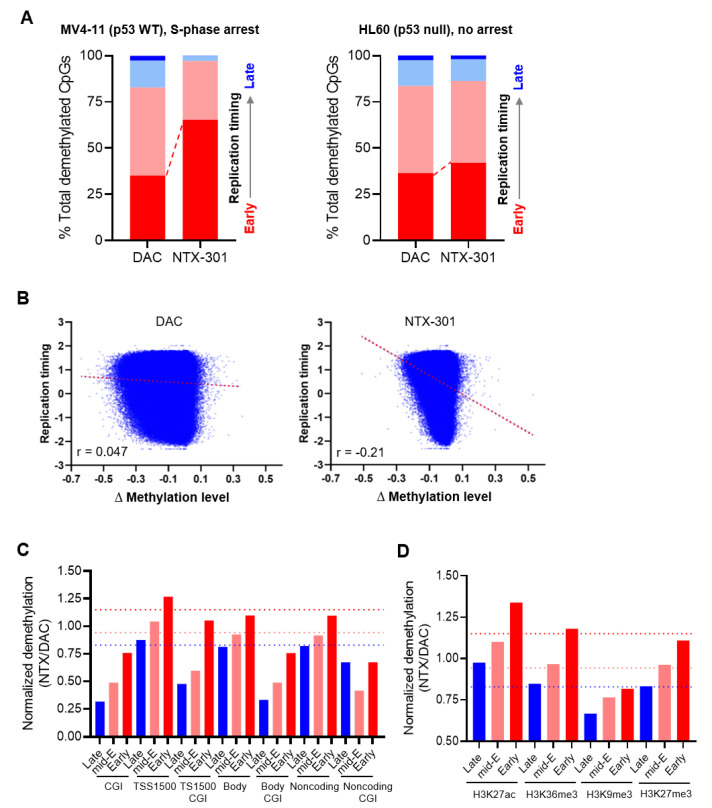
NTX-301 promoted characteristic demethylation patterns. (**A**), Plots showing the distribution of CpGs that were demethylated after treatment with NTX-301 or DAC in four genomic regions annotated by replication timing from early (red) to late (blue) in MV4-11 (left) and HL-60 (right) cells. (**B**), Scatter plots showing differential methylation levels (∆methylation level = on-treatment level − basal level) plotted against replication timing in MV4-11 cells. Correlation coefficients (r) are shown. (**C**,**D**), Bar plots showing the normalized ratio of NTX-301-mediated demethylation to DAC-mediated demethylation in several genomic (**C**) and chromatin (**D**) features. The red, pink, and blue dotted lines denote the normalized baselines of the demethylation ratio in genomic regions with early, mid-early (mid-E), and late replication timing, respectively. CGI, CpG island; TSS1500, the region 1500 bp upstream of the transcription start site.

## Data Availability

All data are available from the NCBI GEO Datasets (https://www.ncbi.nlm.nih.gov/geo/) via the accession numbers GSE188392, GSE187285, and GSE187293.

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
