# Peer review of "Integrative Analyses Reveal the Anticancer Mechanisms and Sensitivity Markers of the Next-Generation Hypomethylating Agent NTX-301"

_cancers, 2023, doi:10.3390/cancers15061737_

Round 1

Reviewer 1 Report (Previous Reviewer 1)

The Authors have met all my remarks  concerning the formal and experimental matters pertaining to the manuscript.

As a minor issue: in lines 237 and 504 the phrase " when treated" should be substituted by "when applied". The sentence in lines 507-509 should be rephrased for the same reason.

Reviewer 2 Report (Previous Reviewer 2)

The authors addressed my points of criticism satisfactorily.

This manuscript is a resubmission of an earlier submission. The following is a list of the peer review reports and author responses from that submission.

Round 1

Reviewer 1 Report

The manuscript by Lim et al. examines the effect of NTX-301, a novel demethylating agent, on the  viability,  apoptosis and differentiation of several AML cell lines, and its efficacy in inhibiting tumor growth in NOD/SCID mice. Moreover, it also analyses changes in the transcriptome and the extent and specificity of DNA demethylation evoked by NTX-301. Most of the presented analyses compare the efficacy of NTX-301 with other hypomethylating agents (HMAs) i.e., decitabine and 5-azacytidine. Also, by using integrative analysis, the Authors expand some of their observations to 199 other cancer cell lines. This is an extensive and detailed study documenting higher efficacy of NTX-301, revealing its mechanism of action (see below) and intricacies of its demethylating activity. I have several comments:

Major:

This manuscript is an extension/continuation of an earlier paper by the Authors (Lim et al.,  Blood Cancer J. 2022, 12(4): 57. I have some doubts about managing the experimental data between these two manuscripts. Namely, the major finding that NTX-301 treatment upregulated genes associated with DDR, p53 pathway, apoptosis and immune response has been reported in the earlier publication. However, although the Authors cite this publication in the current paper (position 11), they do not do it in the context of this finding neither when describing the results (lines 252-262) or in the Discussion. Also, some graphs (upper panels in Fig. 3A and in Fig. 4B,C and D) seem to be the same as in their previous publication (Fig 5S). While the rationale of presenting again the control panel in Fig. 3A has been justified in the text (lines 223-225) there is no comment regarding the other graphs.

Minor:

1. Line 230. The concentrations of NTX-301 used here in comparison with a previous study should be given rather as “times/fold lower” than as a percentage, especially that 0.45 mg/kg corresponds to 77.5 and not to 80% of 2.0 mg/kg.

2. Line 304-306 and Fig. 5C. There seem to be no obvious correlation between IC50 and expression for Bak.

Reviewer 2 Report

The manuscript presented by Lim and colleagues investigates the mechanism of action of NTX-301, a nucleoside DNMT inhibitor, which is an orally active compound. The authors suggest that this inhibitor is superior compared to decitabine in terms of its in vitro and in vivo efficacy. It is important to consider that DNMT inhibitors show cytotoxic effects when used in high doses. Enhanced effects were observed with low doses, which results in optimal DNMT inhibition and thus passive demethylation. This fact should also be considered by the authors and discussed in their manuscript. Although the authors used low dose treatments for their in vivo experiments, it is somehow opaque, which experimental settings (doses and time frame) were used in the different in vitro experiments. As mentioned above, the DNMT1 inhibition results in passive demethylation of methylated CpGs, which can only be assessed after several rounds of replication. It is also unclear, how NTX-301 and DAC concentrations were chosen. To compare the efficacy of the two drugs, drug doses should be normalized to their IC50 concentrations.  

·      In the introduction, the authors should also mention the recently described novel enzymatic DNMT inhibitor GSK3685032, which is a non-nucleoside analogue.

·      The methods section lacks detail as to the specific treatment time and doses and in silico analyses. 

·      The IC50 curves the DAC treatment curves are too flat, higher doses should be used to be able to compare the IC50 values, why were some treated for 1/3 days other cell lines for 2/4 days?

·      Fig. 2A and supplementary Figure S1 are redundant except for the quantification. This could be included in the main figure.

·      Fig.2B-D: How was the dose for the treatment selected? The quality of FACS data should be improved. Some images appear squeezed. Axes of PI and specific AB stainings are switched in the individual measurements. The cell cycle FACS histograms should include sub-G1 regions. It is surprising that the DMSO control cells show high levels of G2/M phase after 72h (supp. Fig. 2). It is unclear how the quantification of the individual cell cycle phases was done (suppl. Fig.2 lower panel), since PI cannot distinguish G2 from M phase. 

·      Fig.2E please include a representative image of DAC treated cells

·      It is unclear why the authors used AZA instead of DAC for their in vivo treatments. 

·      Details on the RNA-seq analysis such as treatment dose and schedule are missing. Parts of the RNA-seq data (Fig.4) were already published in their recent paper Blood Cancer J. 2022 Apr; 12(4): 57.

·      Please provide more details on the CCLE data analyses. Which cell lines and which inhibitors were used for the analyses? Please list the datasets used and include respective references.

·      Details on DNA methylation analyses are missing

·      It would be interesting to investigate whether NTX-301 also induces viral mimicry.